green chemistry/materials science/environmental science

spent lithium-ion batteries, metal recovery, maleic acid, glycolic acid, acetoacetic acid

**Authors for correspondence:**
Qing Huang
e-mail: huangqing3121@sina.com
Yuefeng Su
e-mail: suyuefeng@bit.edu.cn.

This article has been edited by the Royal Society of Chemistry, including the commissioning, peer review process and editorial aspects up to the point of acceptance.

# Maleic, glycolic and acetoacetic acids-leaching for recovery of valuable metals from spent lithium-ion batteries: leaching parameters, thermodynamics and kinetics

Borui Liu[1], Qing Huang[1], Yuefeng Su[1], Liuye Sun[1], Tong Wu[1], Guange Wang[1], Ryan M. Kelly[2] and Feng Wu[1]

[1]School of Materials Science and Engineering, Beijing Institute of Technology, Beijing 100081, People's Republic of China
[2]Rykell Scientific Editorial, LLC, Los Angeles, CA, USA

QH, 0000-0001-5309-9276; YS, 0000-0002-5144-2832

Environmentally friendly acid-leaching processes with three organic acids (maleic, glycolic and acetoacetic) were developed to recover valuable metals from the cathodic material of spent lithium-ion batteries ($LiCoO_2$). The leaching efficiencies of Li and Co by the maleic acid were 99.58% and 98.77%, respectively. The leaching efficiencies of Li and Co by the glycolic acid were 98.54% and 97.83%, while those by the acetoacetic acid were 98.62% and 97.99%, respectively. The optimal acid concentration for the maleic acid-, glycolic acid- and acetoacetic acid-leaching processes were 1, 2 and 1.5 mol l$^{-1}$, respectively, while their optimal $H_2O_2$ concentrations were 1.5, 2 and 1.5 vol%, respectively. The optimal solid/liquid ratio, temperature and reaction time for the leaching process of the three organic acids was the same (10 g l$^{-1}$, 70°C, 60 min). The thermodynamic formation energy of the leaching products and the Gibbs free energy of the leaching reactions were calculated, and the kinetic study showed that the leaching processes fit well with the shrinking-core model. Based on the comparison in the leaching parameters, the efficacy and availability of the three acids is as follows: maleic acid > acetoacetic acid > glycolic acid.

# 1. Introduction

Lithium-ion batteries (LIBs) have been widely adopted in electronic devices and power tools because of their higher energy density, lower size/weight, as well as lower memory effects [1–4]. Therefore, the production of LIBs has increased gradually in the last decades by virtue of their wide application as a power supply. By the year 2020, it is predicted that the number of LIBs in the world will exceed more than 25 billion units [5,6]. The most common active components of the cathodes of LIBs are lithium transition metal oxides, such as $LiCoO_2$, $LiMn_2O_4$, $LiNiO_2$ and $LiCo_xMnyNi_zO_2$. It is understood that the cathodic materials from LIBs contain several valuable elements, such as Li, Co, Ni and Mn, which are important resources and raw materials for industrial manufacturers [6,7]. Therefore, as the lifetime of each battery expires, the valuable metals in the cathodic material from spent LIBs need to be recycled in order to reduce the waste of these valuable resources.

The two main methods for metal recovery from cathodic material of spent LIBs include pyrometallurgical [8] and hydrometallurgical [9,10]. Hydrometallurgy is a well-established process for the separation and recovery of metal ions. Unlike the pyrometallurgical technique, the hydrometallurgical technique can achieve complete recovery of metals with high purity with low energy consumption, the minimization of wastewater and no gases emission [11]. Generally, the hydrometallurgical recycling process includes pretreatment, acid-leaching and reclamation [5,12,13]. For this process, acid-leaching is an important step in transforming valuable metals from the solid lithium transition metal oxides to soluble ions under the action of acids. There are a lot of inorganic and organic acids which can be effective leaching agents. Inorganic acids like HCl [14], $H_2SO_4$ [15,16], $HNO_3$ [17] and phosphoric acid [18–20] have been proved effective for the acid-leaching process. However, inorganic acid may lead to serious secondary pollution [21]. Therefore, natural and easily degradable organic acids have been employed as leaching agents in previous studies, which can help to minimize the environmental impact of the acid-leaching process [22]. In previous studies, it has been proved that organic acids such as acetic acid [23], citric acid [18,24–26], D, L-malic acid [27,28], ascorbic acid [29], succinic acid [30], oxalic acid [31], trichloroacetic acid [32], iminodiacetic acid [33], tartaric acid [34,35], benzenesulfonic acid [36] and lactic acid [22] can achieve high leaching efficiencies of valuable metals from spent LIBs. Moreover, there is another advantage of organic acids as the leaching agents. The organic acids leachate of spent LIBs can be directly used as the material to resynthesize cathodic material of LIBs through the sol–gel method, without any intermediate step or loss of materials [5,22,37]. Therefore, a closed-loop process of recovery-resynthesis can be achieved when organic acids are employed as the leaching agents. This process can not only reduce the recovery cost but also minimize the emission of TOC and other organic hazardous substances.

There are other natural organic acids remaining which have the potential to be effective agents for the acid-leaching process, but direct research on these acids is sparse. For example, maleic acid, glycolic acid and acetoacetic acid are three organic acids naturally existing in plants and animals, which are all environmentally friendly agents. In the present study, the three organic acids were employed as the acid-leaching agents in the metal recycling process of cathodic material in spent LIBs. The optimal leaching condition, by evaluating acid concentration, reductant concentration, solid/liquid ratio (S/L), temperature, and reaction time of the three organic acids were studied. Additionally, thermodynamic and kinetic studies were commenced in order to investigate the reaction mechanisms. Based on parameters in the acid-leaching process, as well as the thermodynamic and kinetic properties, the availability of the three organic acids were compared and ranked.

# 2. Material and methods

## 2.1. Materials and reagents

Spent LIBs were collected from the Zhongguancun Electronic Market located in Beijing, China. NaCl was applied to completely discharge the spent LIBs. NaOH was used to dissolve the Al foil for separating the cathodic materials. $HNO_3$ and HCl were applied to digest the spent cathodic materials and leaching residues. The maleic, glycolic and acetoacetic acids were applied as the leaching agent, while $H_2O_2$ was used as a reductant in the acid-leaching process. All of the reagents in the experiment were of analytical grade, and all solutions were prepared with distilled water.

## 2.2. Experimental procedure

### 2.2.1. Pretreatment process

Spent LIBs were first soaked in the NaCl solution to be completely discharged and avoid potential dangers during the subsequent recycling process [38]. Then, the discharged LIBs were manually dismantled and their plastic and steel cases were removed. The cathode foil was separated from the LIBs and cut into pieces of $1 \times 1$ cm$^2$. The cathode pieces were then introduced into the NaOH solution to dissolve the Al foil and separate the active cathodic material. This separated cathodic material was dried at 70°C.

The thermal behaviour of the spent cathodic material after reaction with the NaOH solution was investigated by thermogravimetric analysis and differential scanning calorimetry (TG-DSC). TG-DSC were performed in air atmosphere at a heating rate of 10°C min$^{-1}$, the results of which are shown in electronic supplementary material, figure S1. The TG curve identifies that there were three weight-loss stages. At the stage of 28 to 384°C, the weight-loss rate was 1.42%, representing the loss of bound water. At the stage of 384 to 848.5°C, the weight-loss rate was 3.30%, representing the burning of acetylene black and decomposition of poly(vinylidene fluoride) (PVDF). At the stage of 848.5 to 1000°C, the weight-loss rate was 4.67%. According to the DSC curve, the endothermic DSC peak at 475°C represented the loss of PVDF and acetylene black, while the exothermic peak at 848.5°C represented the phase change of the lithium transition oxides and the loss of lithium [22,27]. Based on the TG-DSC curves, there was no significant weight loss in the range of 600–800°C, while there was a phase change in the active cathodic material at 848.5°C. Therefore, in order to remove the acetylene black and PVDF, the spent cathode material after reaction with the NaOH solution was calcined at 800°C for 5 h in a muffle furnace, in which pure active cathodic material was obtained. Thenceforth, the obtained material was grounded into powder with a planetary ball mill for about 30 min to increase the subsequent leaching efficiency.

### 2.2.2. Characterization of the cathodic material

The X-ray diffraction (XRD) was used to determine the crystal structures of the pure active cathodic material before acid-leaching, whose results are shown in electronic supplementary material, figure S2(a). It is discovered that nearly all the diffraction peaks in the XRD patterns were indexed to LiCoO$_2$, and a small amount of Co$_3$O$_4$ can be detected in the active cathodic material [29]. With respect to the leaching residues from the three organic acids, the XRD patterns show that the main component of the residues was Co$_3$O$_4$ (electronic supplementary material, figure S2(a)).

The surface appearance of the pure active cathodic material was investigated using field-emission scanning electron microscopy (FESEM), while the energy dispersive spectrometer (EDS) was applied to analyse the surface element component of the material. The FESEM image of the active cathodic material before leaching is shown in electronic supplementary material, figure S2(b). According to the image, the active cathodic material showed irregular morphologies with a particle size ranging from 10 to 40 μm. After leaching by the three organic acids, the size of the particles was much smaller than the active cathodic material before leaching (electronic supplementary material, figure S2c–e). Moreover, from the EDS results (electronic supplementary material, table S1), the active cathodic material and residues were composed of Co and O species, while Li failed to be detected due to the low energy density. No other elements were detected by the EDS. Furthermore, the EDS results showed that the mole ratio of Co and O on the particle of active cathodic material was approximately 1 : 2, while the mole ratio of Co and O on the particle of the leaching residue from the three organic acids was approximately 3 : 4.

To further determine the element component in the active cathodic material and leaching residues, the powder was digested with aqua regia (HCl : HNO$_3$ = 3 : 1, in volume) and hydrogen peroxide (H$_2$O$_2$) [39], and metal concentrations were determined using the inductively coupled plasma-optical emission spectroscopy (ICP-OES). The results showed that in the active cathodic material, the weight rate of Li was 7.60%, while that of Co was 61.48%. The weight rates of Co in the leaching residues from maleic, glycolic and acetoacetic acids were 73.42%, 73.20% and 73.35%, respectively. Based on the XRD, EDS and ICP-OES results, the component of the active cathodic material before leaching was LiCoO$_2$, while that of the leaching residues from the three organic acids was Co$_3$O$_4$.

### 2.2.3. Acid-leaching process

Based on the characterization results, the active cathodic material from the spent LIBs was LiCoO$_2$, and the recyclable metals in the material were Li and Co. The acid-leaching process can be represented by

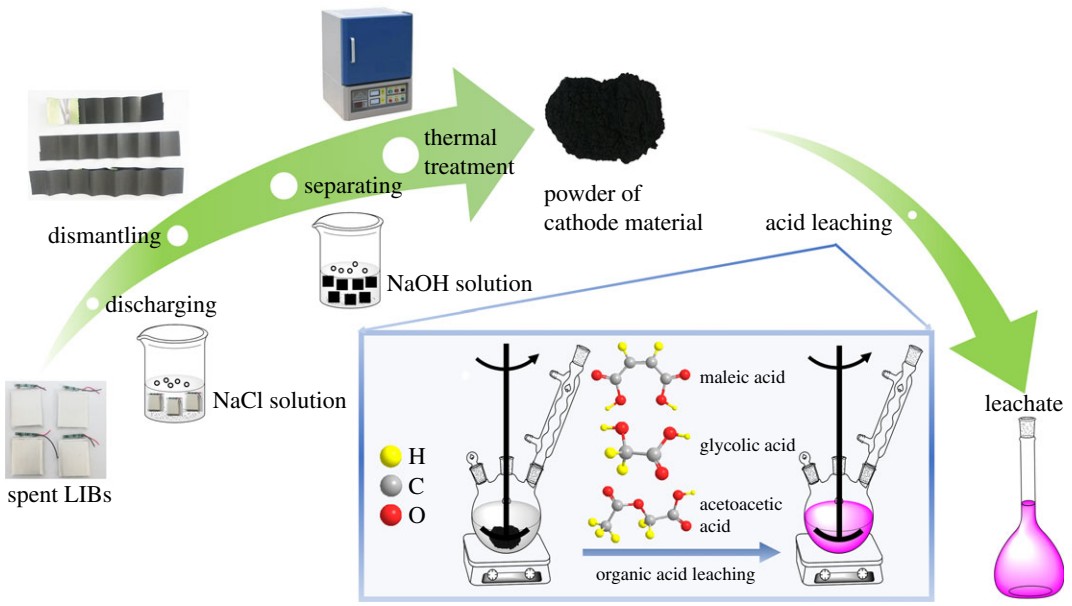

**Figure 1.** Scheme of the pretreatment and acid-leaching process in the experiment.

equation (2.1).

$$6H^+ + 2LiCoO_2 + H_2O_2 \rightarrow 2Li^+ + 2Co^{2+} + 4H_2O + O_2. \tag{2.1}$$

The dissociation constant of an organic acid is an important factor affecting the $H^+$ concentration in the solution, as well as the leaching efficiency. The dissociation of three organic acids in the present study can be expressed in equations (2.2)–(2.5).

$$\text{maleic acid:}\quad C_4H_4O_4 \rightarrow C_4H_3O_4^- + H^+ \quad K_{a1} = 1.42 \times 10^{-2}, \tag{2.2}$$

$$C_4H_3O_4^- \rightarrow C_4H_2O_4^- + H^+ \quad K_{a2} = 8.57 \times 10^{-7}, \tag{2.3}$$

$$\text{glycolic acid:}\quad C_2H_4O_3 \rightarrow C_2H_3O_3^- + H^+ \quad K_a = 1.48 \times 10^{-4} \tag{2.4}$$

and $\quad$ acetoacetic acid: $\quad C_4H_6O_4 \rightarrow C_4H_5O_4^- + H^+ \quad K_a = 2.62 \times 10^{-4}. \tag{2.5}$

The dissociation constants of the three organic acids are comparable to those of the other available organic acids in previous studies [3,27,40]. Therefore, the maleic, glycolic and acetoacetic acids could be potential acid-leaching agents for active cathodic materials. Also, because Co(III) is not leachable, the reductant is necessary to reduce Co(III) into Co(II). According to the standard electrode potential, $\Phi^{\Theta}(Co^{3+}/Co^{2+}) = 1.83$ and $\Phi^{\Theta}(O^2/H_2O_2) = 0.699$ [27,41]. Hence, $H_2O_2$ in the acid-leaching process is a potential and available reductant in the process. The scheme of the pretreatment and acid-leaching process is shown in figure 1.

To conduct the acid-leaching experiment, the prepared powder was introduced into a 150 ml three-necked, round-bottomed, thermostatic Pyrex reactor. The reactor was placed into a water bath to obtain a constant reaction temperature. The reactor was assembled with a mechanical stirrer and a condenser pipe to accelerate the reaction and avoid evaporation, respectively. Thereafter, the organic acid and $H_2O_2$ solution were introduced into the reactor and the mixture was then stirred at 300 r.p.m. After leaching, the leachate was separated with the residue by centrifugation. In order to determine the leaching efficiency, the leachate was firstly diluted to 200 ml with $HNO_3$ solution (1%), and then the concentrations of Co and Li were determined with ICP-OES. The leaching efficiency of the metals was calculated with equation (2.6):

$$X = \frac{A_e}{A_0}, \tag{2.6}$$

where $X$ represents the leaching efficiency, $A_e$ represents the amount of metals in the leachate (milligram), and $A_0$ represents the amount of metals in the pure active cathodic material before acid-leaching (milligram).

Based on the experimental method above mentioned, an orthogonal experiment was conducted to investigate the order of the acid-leaching parameters, which affected the leaching efficiency. The results

are displayed in electronic supplementary material, table S2–S5. The existence of Li and Co ions illustrates that metal ions could be leached from active cathodic material by the three organic acids (electronic supplementary material, table S2–S4). The leaching efficiency of Li and Co varied with the five leaching parameters. The results of range analysis (electronic supplementary material, table S5) indicated that the five leaching parameters affected the leaching efficiency of the three organic acids in the following order: acid concentration > $H_2O_2$ concentration > solid/liquid ratio > temperature > reaction time.

In the subsequent acid-leaching process, the following parameters were studied to obtain the optimal acid-leaching condition: (i) organic acid concentration (0.5–2.5 mol l$^{-1}$), (ii) $H_2O_2$ concentration (0.5–2.5 vol%), (iii) S/L ratio (5–40 g l$^{-1}$), (iv) temperature (50–90°C) and (v) reaction time (30–70 min). The thermodynamic analysis for the acid-leaching reactions was conducted, including formation energy of the possible leaching products (calculated using Materials Studio software), as well as the Gibbs free energy of the reactions ($\Delta_r G_m$). A kinetics study of the leaching process was conducted under the following conditions: (i) temperature (50–90°C) and (ii) reaction time (10–70 min), while the other parameters were maintained at the above mentioned optimal acid-leaching condition.

# 3. Results and discussion

## 3.1. Effects of acid-leaching parameters on the leaching efficiency

### 3.1.1. Optimal acid-leaching parameters of maleic acid

In order to recognize the optimal acid concentration of the maleic acid-leaching process, the maleic acid concentration was varied from 0.5 to 2.5 mol l$^{-1}$, while the other parameters were maintained at the following levels: $H_2O_2$ concentration = 1.5 vol%, S/L ratio = 10 g l$^{-1}$, temperature = 70°C and reaction time = 60 min. The effect of maleic acid concentration on the leaching efficiency is shown in figure 2a. When the maleic acid concentration was 0.5 mol l$^{-1}$, the leaching efficiencies of Li and Co were 89.98 and 88.50%, respectively. However, when the maleic acid concentration increased to 1 mol l$^{-1}$, the leaching efficiencies of the two metals both reached 99.54%. With a further increase in the maleic acid concentration from 1.5 to 2.5 mol l$^{-1}$, the leaching efficiencies did not increase significantly. The reason for this phenomenon might be influenced by different rate-controlling factors at different acid concentrations. As the acid concentration increased from 0.5 to 1 mol l$^{-1}$, the rate-controlling factor was the chemical reaction rate which increased with the acid concentration. However, when the acid concentration was larger than 1 mol l$^{-1}$, the rate-controlling factor changed into the rate of transferring ions, and the leaching efficiency failed to further increase [41]. Therefore, the optimal maleic acid concentration in the acid-leaching process was 1 mol l$^{-1}$.

To identify the optimal $H_2O_2$ concentration of the maleic acid-leaching process, the $H_2O_2$ concentration varied from 0.5 to 2.5 vol%, while the other parameters were maintained at the following levels: maleic acid concentration = 1 mol l$^{-1}$, S/L ratio = 10 g l$^{-1}$, temperature = 70°C and reaction time = 60 min. Figure 2b shows the impact of $H_2O_2$ concentration on the Li and Co leaching efficiency by maleic acid. As the $H_2O_2$ concentration increased from 0.5 to 1.5 vol%, the leaching efficiency of Li increased gradually from 76.08% to 99.39%, while that of Co increased from 73.47 to 98.26% (figure 2b). As the $H_2O_2$ concentration further rose to 2 and 2.5 vol%, the leaching efficiency of Li and Co was maintained at a stable level. The reason for this phenomenon might be that Co of $LiCoO_2$ is almost completely leached in the presence of 1.5 vol% $H_2O_2$ and Co of $Co_3O_4$ present in the leaching residue is not leachable. Therefore, as the $H_2O_2$ concentration further increased, there was no significant increase in the leaching efficiencies of Co and Li. Thence, the optimal $H_2O_2$ concentration for the maleic acid-leaching process was 1.5 vol%.

Figure 2c shows the impact of S/L ratio on the Li and Co leaching efficiency by maleic acid. The optimal S/L ratio was investigated by varying the parameter from 5 to 40 g l$^{-1}$, while other parameters were maintained at the following levels: maleic acid concentration = 1 mol l$^{-1}$, $H_2O_2$ concentration = 1.5 vol%, temperature = 70°C and reaction time = 60 min. When the S/L ratio was 5 and 10 g l$^{-1}$, the Li leaching efficiency was 99.28%, while the Co leaching efficiency was 98.27% (figure 2c). However, with a further increase in the S/L ratio (20 to 40 g l$^{-1}$), the leaching efficiency of the two metals decreased significantly. Therefore, the optimal S/L ratio for the maleic acid-leaching process was 10 g l$^{-1}$. This phenomenon indicated the shortage of the acid-leaching technology for the recovery of metals from spent LIBs when the S/L ratio (pulp density) was high.

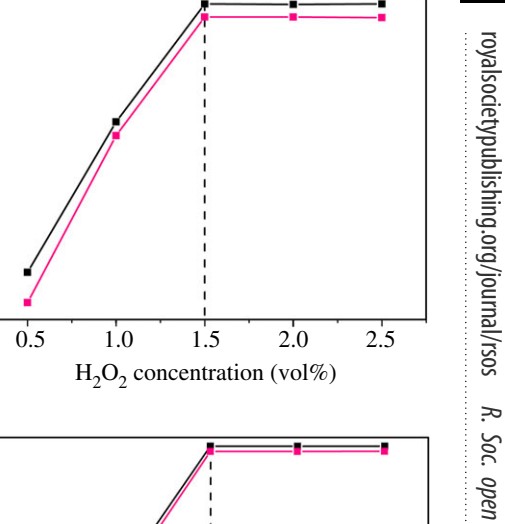

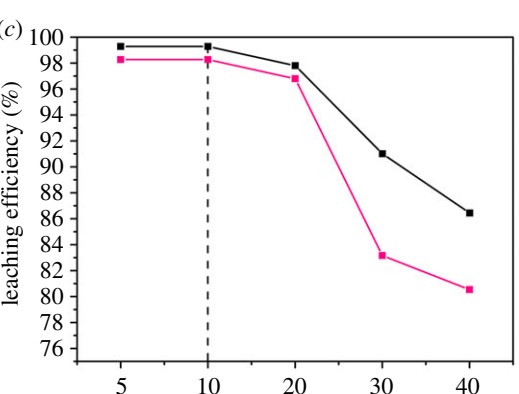

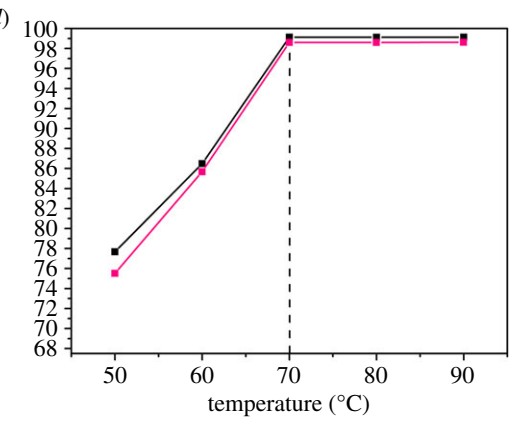

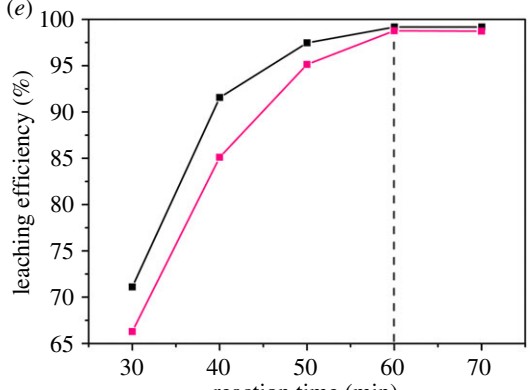

**Figure 2.** Effects of the acid-leaching parameters on the maleic acid-leaching efficiency. Maleic acid concentration (*a*); H$_2$O$_2$ concentration (*b*); S/L ratio (*c*); temperature (*d*); reaction time (*e*).

To study the optimal temperature for the maleic acid-leaching process, the temperature was varied from 50 to 90°C, while the other parameters were maintained at the following levels: maleic acid concentration = 1 mol l$^{-1}$, H$_2$O$_2$ concentration = 1.5 vol%, S/L ratio = 10 g l$^{-1}$ and reaction time = 60 min. Figure 2*d* shows the effect of temperature on the Li and Co leaching efficiency by maleic acid. As the temperature increased from 50 to 70°C, the leaching efficiency of Li increased gradually from 77.66 to 99.13%, while that of Co increased from 75.52 to 98.60%. As the temperature further reached 80 and 90°C, the leaching efficiency of Li and Co was maintained at a stable level. The reason for this response might be that Co of LiCoO$_2$ was almost completely leached when the temperature was 70°C, while Co of Co$_3$O$_4$ present in the residue was not leachable. Therefore, as the temperature further increased, there was no significant increase in the leaching efficiencies of Co and Li. Therefore, the optimal temperature of the maleic acid-leaching process was 70°C.

To examine the optimal reaction time of the maleic acid-leaching process, the parameter was varied from 30 to 70 min, while the other parameters were maintained at the following levels: maleic acid concentration = 1 mol l$^{-1}$, H$_2$O$_2$ concentration = 1.5 vol%, S/L ratio = 10 g l$^{-1}$ and temperature = 70°C.

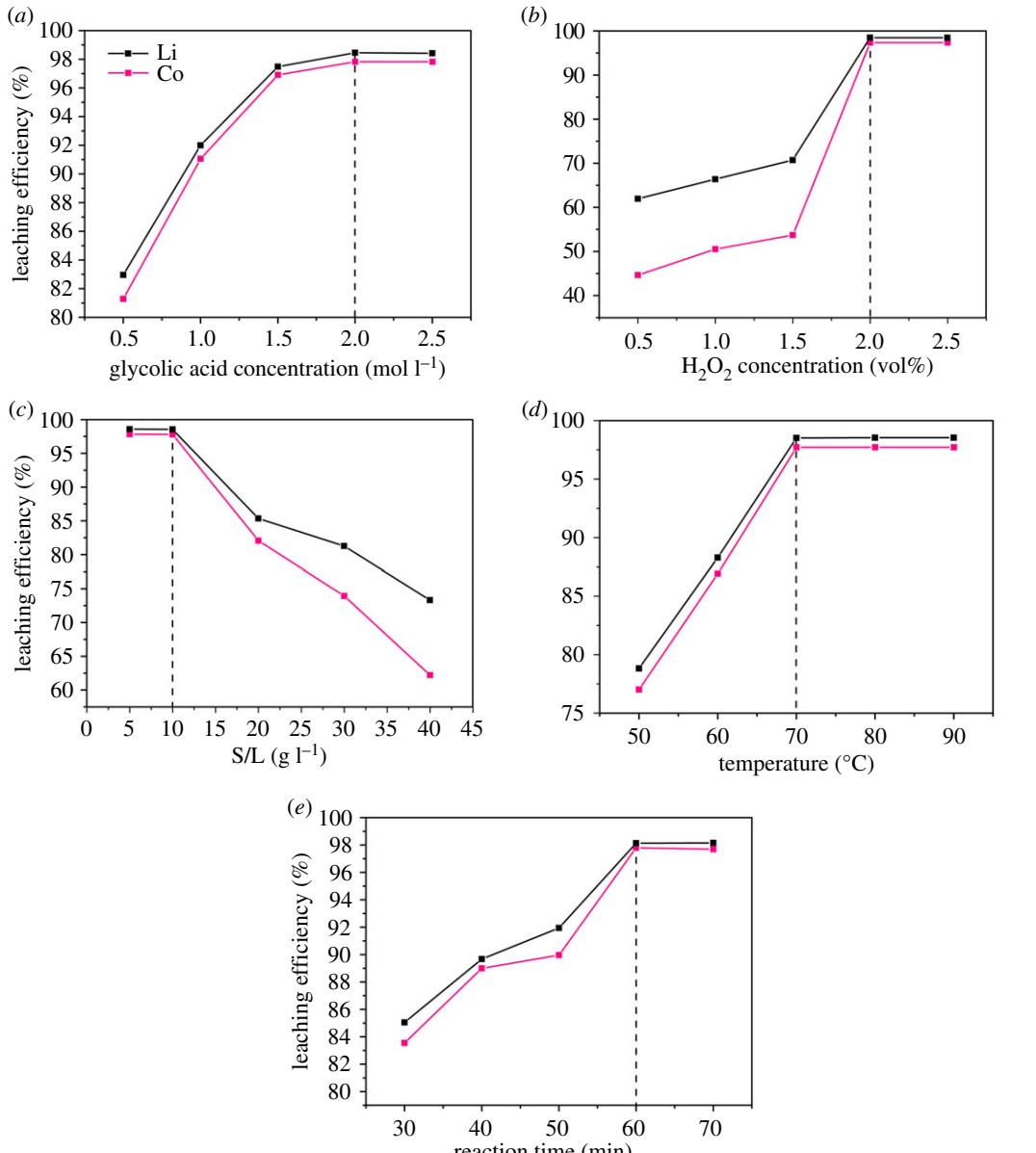

**Figure 3.** Effects of the acid-leaching parameters on the glycolic acid-leaching efficiency. Glycolic acid concentration (a); $H_2O_2$ concentration (b); S/L ratio (c); temperature (d); reaction time (e).

Figure 2e shows the effect of the reaction time on the Li and Co leaching efficiency by maleic acid. The leaching efficiency of Li increased gradually from 71.11 to 99.17%, while that of Co increased from 66.30 to 98.77% as the reaction time was increased from 30 to 60 min. While the reaction time reached 70 min, the leaching efficiency of Li and Co was maintained at a stable level. Therefore, the optimal reaction of the maleic acid-leaching process was 60 min.

### 3.1.2. Optimal acid-leaching parameters of glycolic acid

For recognizing the optimal acid concentration of the glycolic acid-leaching process, the glycolic acid concentration was varied from 0.5 to 2.5 mol l$^{-1}$, while the other parameters were maintained at the following levels: $H_2O_2$ concentration = 2 vol%, S/L ratio = 10 g l$^{-1}$, temperature = 70°C and reaction time = 60 min. The effect of glycolic acid concentration on the leaching efficiency is shown in figure 3a. When the glycolic acid concentration was 0.5 mol l$^{-1}$, the leaching efficiencies of Li and Co were 82.96 and 81.29%, respectively. However, when the glycolic acid concentration increased to 2 mol l$^{-1}$, the leaching efficiencies of the two metals both reached 98.46 and 97.83%, respectively, for Li and Co. When the glycolic acid concentration further increased to 2.5 mol l$^{-1}$, the leaching efficiencies did not increase significantly. Therefore, the optimal glycolic acid concentration in the acid-leaching process was 2 mol l$^{-1}$.

To determine the optimal $H_2O_2$ concentration of the glycolic acid-leaching process, the $H_2O_2$ concentration was varied from 0.5 to 2.5 vol%, while the other parameters were maintained at the following levels: glycolic acid concentration = 2 mol $l^{-1}$, S/L ratio = 10 g $l^{-1}$, temperature = 70°C and reaction time = 60 min. Figure 3b shows the impact of $H_2O_2$ concentration on the Li and Co leaching efficiency by glycolic acid. As the $H_2O_2$ concentration increased from 0.5 to 2 vol%, the leaching efficiency of Li increased gradually from 61.97 to 98.48%, while that of Co increased from 44.66 to 97.36%. As the $H_2O_2$ concentration further rose to 2.5 vol%, the leaching efficiency of Li and Co was maintained at a stable level. The optimal $H_2O_2$ concentration of the glycolic acid-leaching process was identified at 2 vol%.

In figure 3c, the impact of the S/L ratio on the Li and Co leaching efficiency by glycolic acid is shown. In investigating the optimal S/L ratio, the parameter was varied from 5 to 40 g $l^{-1}$, while the other parameters were maintained at the following levels: glycolic acid concentration = 2 mol $l^{-1}$, $H_2O_2$ concentration = 2 vol%, temperature = 70°C and reaction time = 60 min. When the S/L ratio was 5 and 10 g $l^{-1}$, the Li leaching efficiencies were 98.59 and 98.55%, respectively, while the Co leaching efficiencies were 97.84 and 97.81% (figure 3c). However, with a further increase in the S/L ratio from 20 to 40 g $l^{-1}$, the leaching efficiency of the two metals decreased significantly. Therefore, the optimal S/L ratio of the glycolic acid-leaching process was 10 g $l^{-1}$.

To study the optimal temperature for the glycolic acid-leaching process, the temperature was varied from 50 to 90°C, while the other parameters were maintained at the following levels: glycolic acid concentration = 2 mol $l^{-1}$, $H_2O_2$ concentration = 2 vol%, S/L ratio = 10 g $l^{-1}$ and reaction time = 60 min. Figure 3d shows the effect of temperature on the Li and Co leaching efficiency by glycolic acid. While the temperature increased from 50 to 70°C, the leaching efficiency of Li increased gradually from 78.83 to 98.52%, and that of Co increased from 77.01 to 97.72%. As the temperature further reached 80 and 90°C, the leaching efficiency of Li and Co was maintained at a stable level. The optimal temperature of the glycolic acid-leaching process was 70°C.

In order to determine the optimal reaction time of the glycolic acid-leaching process, the parameter was varied from 30 to 70 min, while the other parameters were maintained at the following levels: glycolic acid concentration = 2 mol $l^{-1}$, $H_2O_2$ concentration = 2 vol%, S/L ratio = 10 g $l^{-1}$ and temperature = 70°C. The effect of reaction time on the Li and Co leaching efficiency by glycolic acid is shown in figure 3e. As the reaction time increased from 30 to 60 min, the leaching efficiency of Li increased gradually from 85.05 to 98.13%, while that of Co increased from 83.55 to 97.79%. While the reaction time reached 70 min, the leaching efficiency of Li and Co was maintained at a stable level. Thence, the optimal reaction of the glycolic acid-leaching process was determined to be 60 min.

### 3.1.3. Optimal acid-leaching parameters of acetoacetic acid

The optimal acid concentration of the acetoacetic acid-leaching process was recognized by varying the acetoacetic acid concentration from 0.5 to 2.5 mol $l^{-1}$, while the other parameters were maintained at the following levels: $H_2O_2$ concentration = 1.5 vol%, S/L ratio = 10 g $l^{-1}$, temperature = 70°C and reaction time = 60 min. The effect of acetoacetic acid concentration on the leaching efficiency is shown in figure 4a. When the acetoacetic acid concentration was 0.5 mol $l^{-1}$, the leaching efficiencies of Li and Co were 77.49 and 73.57%, respectively. However, when the acetoacetic acid concentration increased to 1.5 mol $l^{-1}$, the leaching efficiencies of the two metals both reached 98.57 and 97.98%, respectively, for Li and Co. When the acetoacetic acid concentration continued to increase to 2 and 2.5 mol $l^{-1}$, the leaching efficiencies did not increase significantly. It was determined that the optimal acetoacetic acid concentration in the acid-leaching process was 1.5 mol $l^{-1}$.

In identifying the optimal $H_2O_2$ concentration of the acetoacetic acid-leaching process, the $H_2O_2$ concentration was varied from 0.5 to 2.5 vol%, while the other parameters were maintained at the following levels: acetoacetic acid concentration = 1.5 mol $l^{-1}$, S/L ratio = 10 g $l^{-1}$, temperature = 70°C and reaction time = 60 min. The impact of $H_2O_2$ concentration on the Li and Co leaching efficiency by acetoacetic acid is shown in figure 4b. As the $H_2O_2$ concentration increased from 0.5 to 1.5 vol%, the leaching efficiency of Li increased gradually from 89.50 to 98.64%, while that of Co increased from 86.36 to 97.99%. As the $H_2O_2$ concentration further rose to 2 and 2.5 vol%, the leaching efficiency of Li and Co was maintained at a stable level. Therefore, the optimal $H_2O_2$ concentration of the acetoacetic acid-leaching process was 1.5 vol%.

Figure 4c shows the impact of the S/L ratio on the Li and Co leaching efficiency by acetoacetic acid. The optimal S/L ratio was investigated by varying the parameter from 5 to 40 g $l^{-1}$, while the other parameters were maintained at the following levels: acetoacetic acid concentration = 1.5 mol $l^{-1}$, $H_2O_2$ concentration = 1.5 vol%, temperature = 70°C and reaction time = 60 min. When the S/L ratio was 5

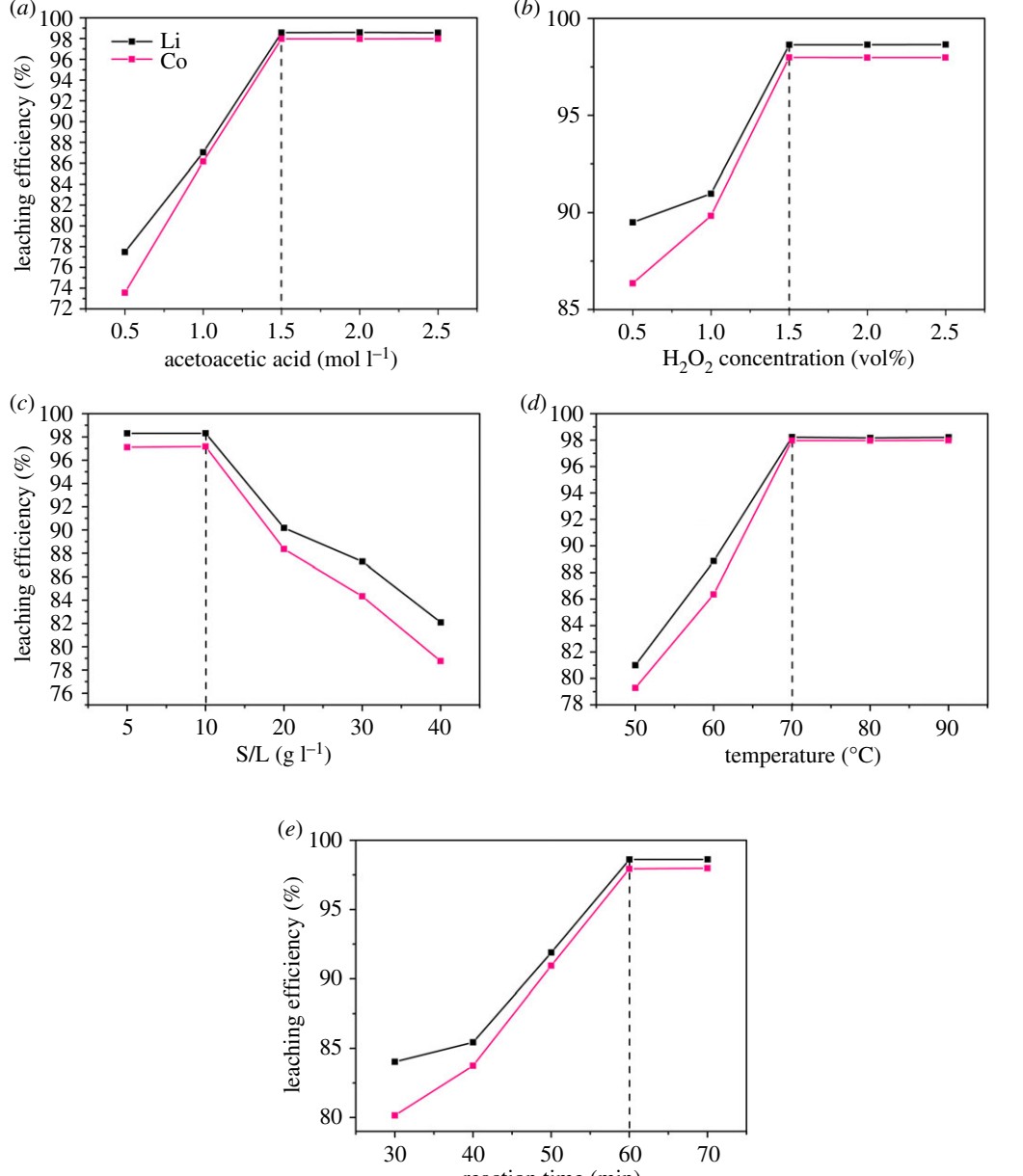

**Figure 4.** Effects of the acid-leaching parameters on the acetoacetic acid-leaching efficiency. Acetoacetic acid concentration (a); H₂O₂ concentration (b); S/L ratio (c); temperature (d); reaction time (e).

and 10 g l$^{-1}$, the Li leaching efficiency was 98.31%, while the Co leaching efficiencies were 97.11 and 97.18% (figure 4c). However, with a further increase in the S/L ratio from 20 to 40 g l$^{-1}$, the leaching efficiency of the two metals decreased significantly. Therefore, the optimal S/L ratio of the acetoacetic acid-leaching process was 10 g l$^{-1}$.

To study the optimal temperature of the acetoacetic acid-leaching process, the temperature was varied from 50 to 90°C, while the other parameters were maintained at the following levels: acetoacetic acid concentration = 1.5 mol l$^{-1}$, H$_2$O$_2$ concentration = 1.5 vol%, S/L ratio = 10 g l$^{-1}$ and reaction time = 60 min. Figure 4d shows the effect of temperature on the Li and Co leaching efficiency by acetoacetic acid. As the temperature increased from 50 to 70°C, the leaching efficiency of Li increased gradually from 80.99 to 98.23%, while that of Co increased from 79.28 to 97.97% (figure 4d). As the temperature continued to reach 80 and 90°C, the leaching efficiency of Li and Co was maintained at a stable level. An optimal temperature for the acetoacetic acid-leaching process was 70°C.

The optimal reaction time of the acetoacetic acid-leaching process was investigated by varying the parameter from 30 to 70 min, while the other parameters were maintained at the following levels: acetoacetic acid concentration = 1.5 mol l$^{-1}$, H$_2$O$_2$ concentration = 1.5 vol%, S/L ratio = 10 g l$^{-1}$ and

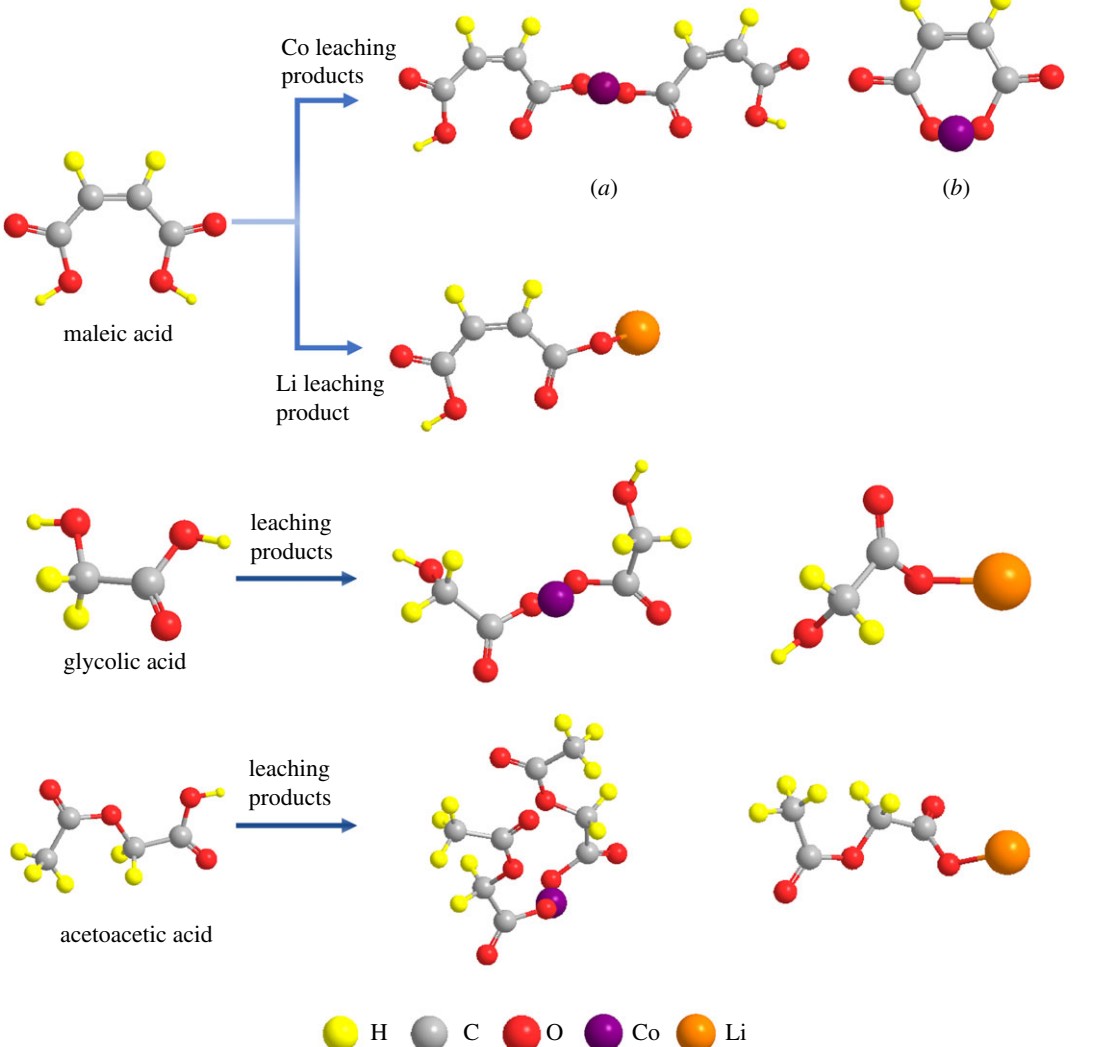

Figure 5. Possible leaching products in the acid-leaching processes of the three organic acids.

temperature = 70°C. The effect of reaction time on the Li and Co leaching efficiency by acetoacetic acid is shown in figure 4e. As the reaction time increased from 30 to 60 min, the leaching efficiency of Li increased gradually from 84.02 to 98.61%, while that of Co increased from 80.16 to 97.93%. As the reaction time further reached 70 min, the leaching efficiency of Li and Co was maintained at a stable level. Thence, the optimal reaction time of the acetoacetic acid-leaching process was 60 min.

In summary, the highest leaching efficiencies of Li and Co by the maleic acid were 99.58% and 98.77%, respectively. The leaching efficiencies of Li and Co by the glycolic acid were 98.54% and 97.83%, while those by the acetoacetic acid were 98.62% and 97.99%, respectively. Based on the XRD patterns of cathodic material and leaching residue, there was $Co_3O_4$ in the cathodic material, which is not leachable. Therefore, the leaching efficiencies of Li were higher than those of Co. The leaching efficiencies of Li by the three organic acids were higher than those by citric acid [24], ascorbic acid [29], oxalic acid [31] and lactic acid [22]. Meanwhile, the leaching efficiencies of Co by the three organic acids in the present study were higher than those by citric acid [24], ascorbic acid [29], oxalic acid [31], trichloroacetic acid [32] and iminodiacetic acid [33]. Therefore, maleic acid, glycolic acid and acetoacetic acid could achieve higher leaching efficiencies of the valuable metals from spent LIBs than some other organic acids.

## 3.2. Thermodynamic analysis of the leaching products

Thermodynamic study is often employed to investigate the mechanisms and behaviour of the chemical reactions by analysing the initial and final state of the reaction system. There were two parameters employed to investigate the thermodynamic mechanisms of the acid-leaching reactions. Firstly, the formation energy of the possible acid-leaching products was calculated, in order to recognize the

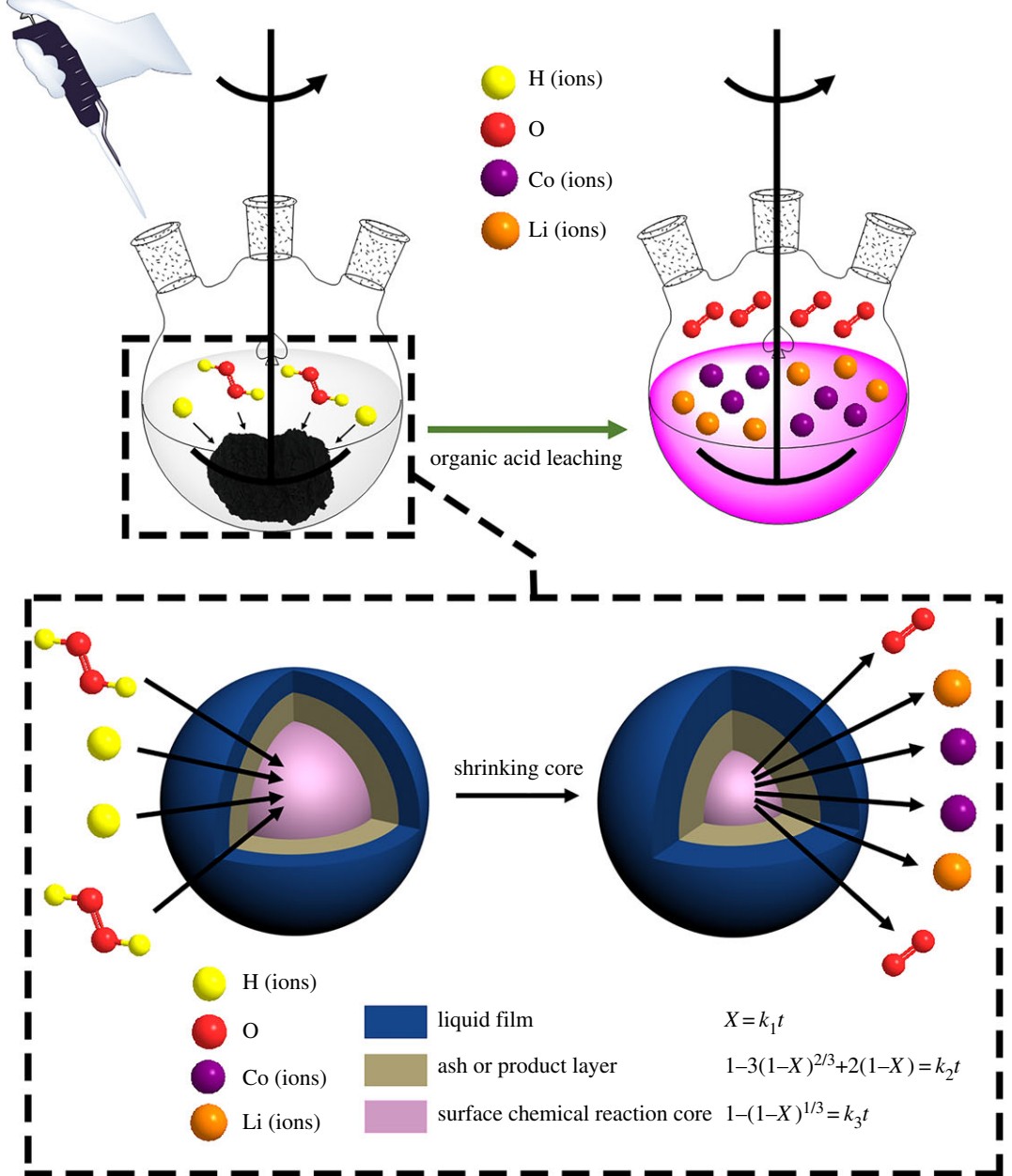

**Figure 6.** Illustration of the kinetic mechanism of the acid-leaching process.

thermodynamic favourable products. Secondly, based on the information about thermodynamic favourable products, the Gibbs free energy ($\Delta_r G_m$) of the acid-leaching reactions was calculated, in order to investigate the final state of the reaction systems, as well as the extent to which leaching reactions proceeded.

In the acid-leaching process, the $LiCoO_2$ cathodic material was dissolved in the three organic acids, and the $H^+$ extracts the $Li^+$ and $Co^{2+}$ within the reduction process by $H_2O_2$. With the assistance of $H_2O_2$, the cobalt in the cathodic material was reduced from Co(III) to Co(II), which makes it more stable and dissolvable in the acid aqueous solution [30]. There are carboxyl groups in the molecules of all the three organic acids, which can be used as ligand to form complexes. Therefore, the leaching products could be generated by the acid radical ions and the $Li^+/Co^{2+}$ through complexation [37]. For the maleic acid, the molecular level contains two carboxyls, and the Co in the leachate was bivalent. Therefore, there were two possible complexation mechanisms for the $Co^{2+}$ in the leachate, leading to two different Co-containing leaching products: $(C_4H_3O_4)_2Co$ and $C_4H_2O_4Co$, which are shown as the product (a) and (b) in figure 5, respectively. To recognize the favourable product during the maleic acid-leaching process, the thermodynamic formation energy of the two possible products at the optimal leaching temperature (70°C) was calculated with a Materials Studio software [32], whose results are shown in electronic supplementary material, table S6. The formation energy of product (a) ($2.214 \times 10^{-19}$ J) was lower than

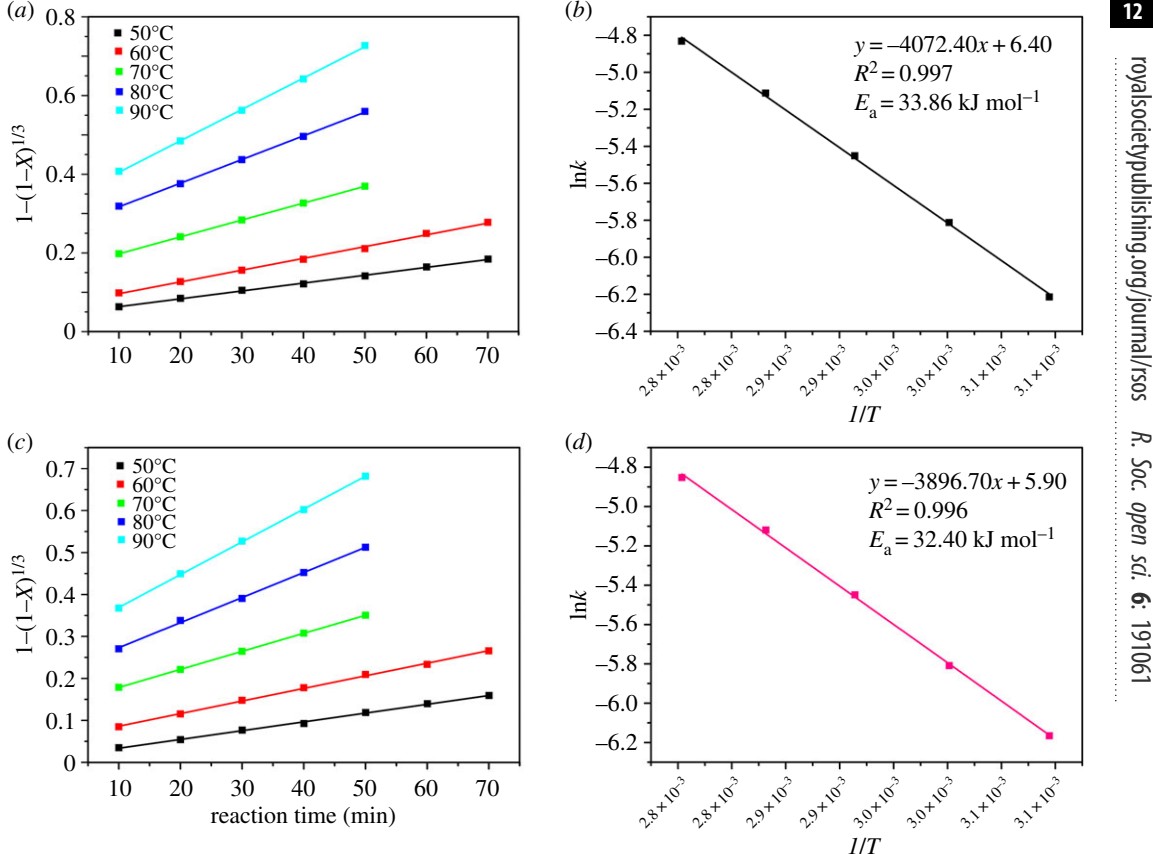

**Figure 7.** Kinetics study of Li (a) and Co (c) in the maleic acid-leaching process (chemical surface reaction model); Arrhenius plots for leaching of Li (b) and Co (d).

that of the product (b) ($5.239 \times 10^{-19}$ J), indicating that the product (a) was thermodynamically favourable. Meanwhile, as can be seen in figure 5, in the maleic acid-leaching process, because Li in the leachate was univalent, there was only one possible Li-containing leaching product: $C_4H_3O_4Li$, whose formation energy was $1.143 \times 10^{-18}$ J (electronic supplementary material, table S6).

The thermodynamic formation energy of the possible leaching products in the glycolic acid- and acetoacetic acid-leaching processes were calculated, whose results are also shown in electronic supplementary material, table S6. There is only one carboxyl group in the molecule of glycolic or acetoacetic acid; therefore, there was only one complexation mechanism for generation of the leaching products. As shown in figure 5, in the glycolic acid-leaching process, the possible leaching products were $(C_2H_3O_3)_2Co$ and $C_2H_3O_3Li$, whose formation energy at the optimal leaching temperature was $2.067 \times 10^{-19}$ and $1.202 \times 10^{-18}$ J, respectively (electronic supplementary material, table S6). In the acetoacetic acid-leaching process, the possible leaching products included $(C_4H_5O_4)_2Co$ and $C_4H_5O_4Li$ (figure 5), whose formation energy at the optimal leaching temperature was $4.543 \times 10^{-18}$ and $2.144 \times 10^{-18}$, respectively (electronic supplementary material, table S6). It can be seen that the formation energy of the acetoacetic acid-leaching products was larger than that of the maleic acid- and glycolic acid-leaching products at 70°C, indicating that the complexation ability of maleic and glycolic acids' radical ions was stronger than acetoacetic acid radical ions. Based on the formation energy calculation results, the acid-leaching processes by the three organic acids could be represented by equations (3.1)–(3.3):

$$6C_4H_4O_4 + 2LiCoO_2(s) + H_2O_2 \rightarrow 2C_4H_3O_4Li + 2(C_4H_3O_4)_2Co + 4H_2O(l) + O_2(g), \quad (3.1)$$

$$6C_2H_4O_3 + 2LiCoO_2(s) + H_2O_2 \rightarrow 2C_2H_3O_3Li + 2(C_2H_3O_3)_2Co + 4H_2O(l) + O_2(g) \quad (3.2)$$

and $$6C_4H_6O_4 + 2LiCoO_2(s) + H_2O_2 \rightarrow 2C_4H_5O_4Li + 2(C_4H_5O_4)_2Co + 4H_2O(l) + O_2(g). \quad (3.3)$$

The Gibbs free energy of the acid-leaching reactions under the optimal leaching conditions was calculated with the following equation:

$$\Delta_r G_m = -RT\ln K, \quad (3.4)$$

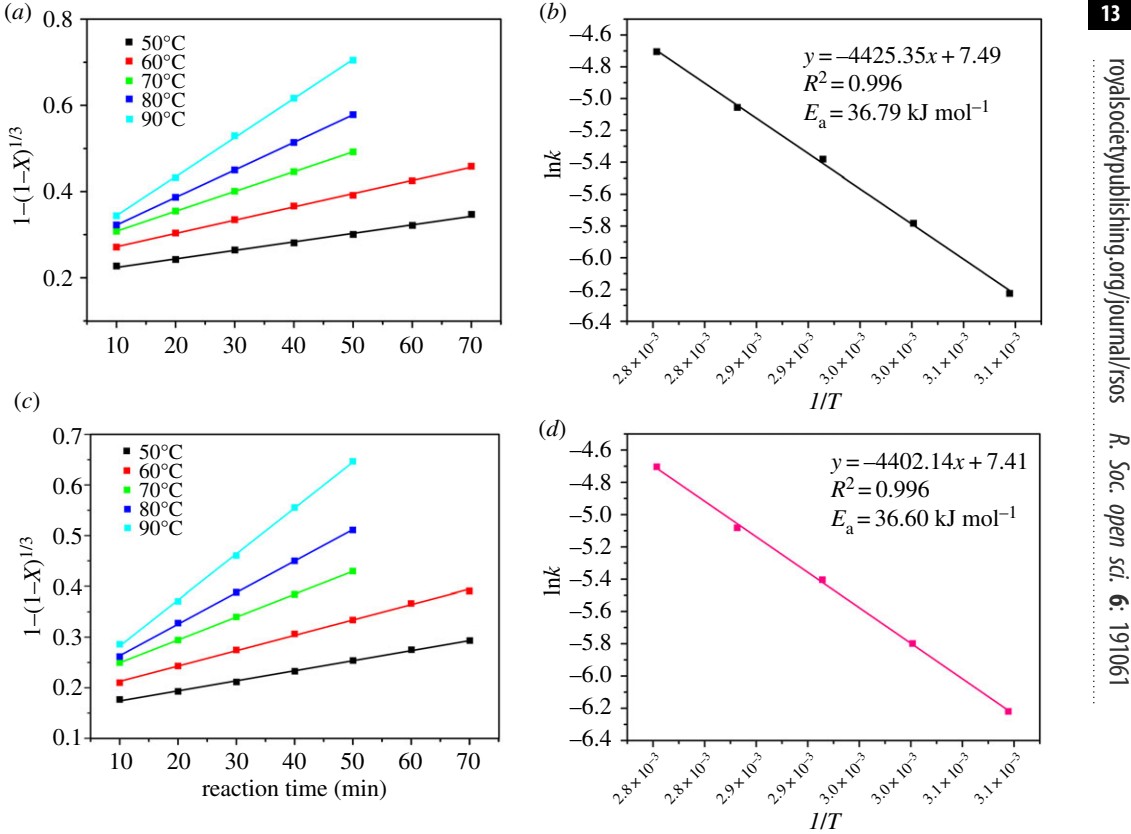

**Figure 8.** Kinetics study of Li (*a*) and Co (*c*) in the glycolic acid-leaching process (chemical surface reaction model); Arrhenius plots for leaching of Li (*b*) and Co (*d*).

where $R$ represents the universal gas constant ($8.314 \, \mathrm{J \, mol^{-1} \, K^{-1}}$), $T$ represents the thermodynamic temperature ($K$), and $K$ represents the equilibrium constants of the leaching reactions. Under the optimal leaching conditions, the reaction temperature was constant (70°C). The equilibrium constants of the leaching reactions can be calculated by the concentrations of leaching products, $H_2O_2$ and organic acids on the final states of the reactions. The concentrations of the above mentioned substances can be obtained based on the stoichiometry in equations (3.1)–(3.3), the highest leaching efficiency, as well as the optimal S/L ratio. The equilibrium constants and Gibbs free energy are shown in electronic supplementary material, table S7. According to the result, under the optimal reaction temperature (70°C), the order of the Gibbs free energy was: $\Delta_r G_{m \, \text{Maleic}} < \Delta_r G_{m \, \text{Acetoacetic}} < \Delta_r G_{m \, \text{Glycolic}}$. The results obtained here illustrate that from the view of thermodynamics, the maleic acid-leaching reaction was the easiest to proceed, while the extent of the glycolic acid-leaching reaction was the least.

## 3.3. Kinetics study on the acid-leaching process

Except for the thermodynamic analysis, the kinetics study was also necessary to investigate the leaching behaviour and feasibility of the leaching reaction by the organic acids. The thermodynamic study was based on the initial and final state of the reaction system, while the kinetic study covered the states of reactants and products in the whole leaching process. The purpose of the kinetic study was to obtain the rate-controlling step, as well as the energy barrier of the reaction. The acid-leaching process from the spent cathodic materials is a typical solid-liquid heterogeneous reaction. The reaction process occurs on the surface of the particles in the reaction system [22]. The shrinking-core models were employed to study the kinetic mechanism of the acid-leaching process [32,37,42]. There are three steps which may be the rate-controlling step, including liquid film diffusion, ash or product layer diffusion and surface chemical reaction [42], which are shown in figure 6. The liquid film diffusion-controlled, ash or product layer diffusion-controlled and surface chemical reaction-controlled process can be represented by the equations shown in figure 6, where $X$ represents the leaching efficiency of Co and Li in the acid-leaching process, and $k_1$ is the liquid film diffusion rate constant ($\mathrm{min^{-1}}$), $k_2$ is the ash or

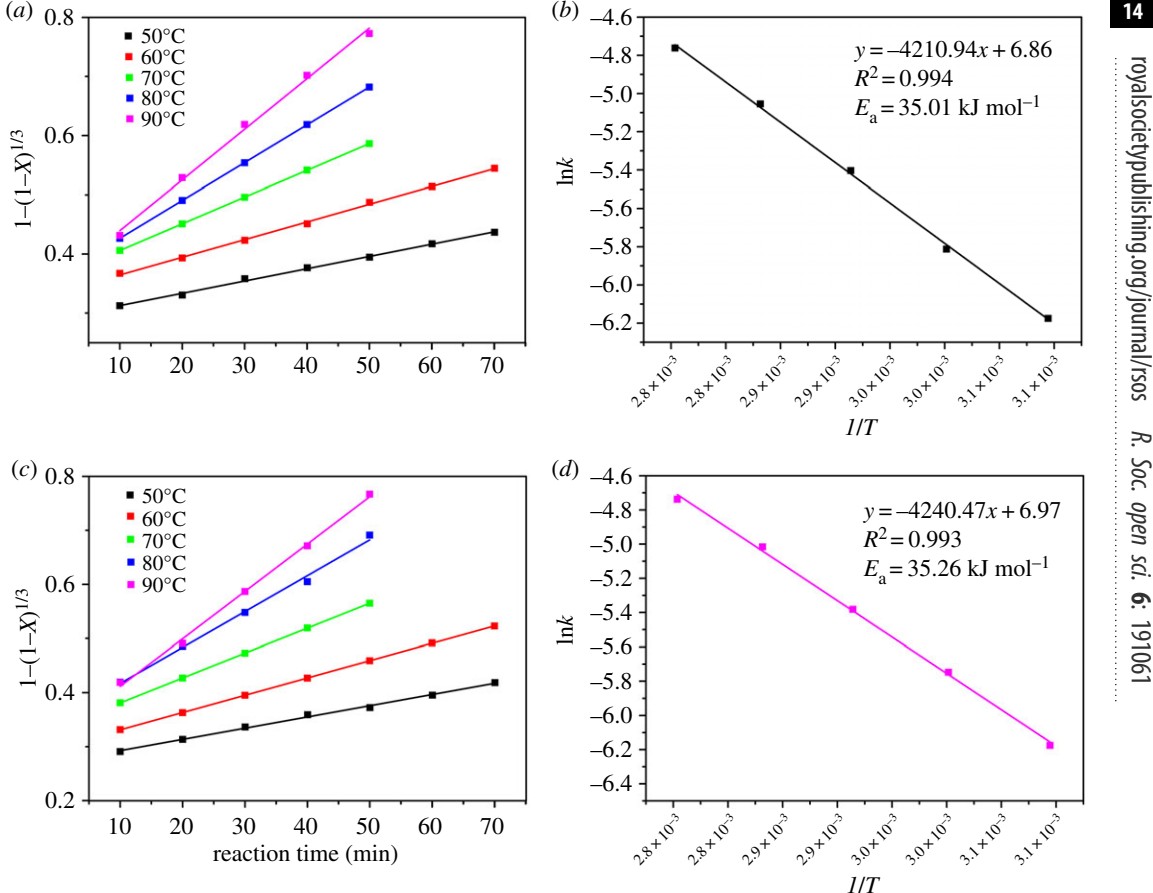

**Figure 9.** Kinetics study of Li (*a*) and Co (*c*) in the acetoacetic acid-leaching process (chemical surface reaction model); Arrhenius plots for leaching of Li (*b*) and Co (*d*).

product layer diffusion rate constant ($min^{-1}$), $k_3$ is the surface chemical reaction rate constant ($min^{-1}$) and $t$ is the reaction time (min). In order to recognize the rate-controlling step, the acid-leaching process involving the three organic acids was performed under different temperatures (50–90°C) and reaction times (10–70 min). The other leaching parameters were maintained at optimal conditions for the three organic acids, respectively. Based on the equation fitting results of the three different steps (figure 6), the rate-controlling step can be recognized. The fitting results of the surface chemical reaction-controlled process for the maleic acid-, glycolic acid- and acetoacetic acid-leaching processes are shown in figures 7–9, and the fitting parameters are shown in electronic supplementary material, table S8. The results in figures 7–9 and electronic supplementary material, table S8 show that the leaching process fit well with the surface chemical reaction-controlled model, and the order of leaching rate of metals by the three organic acids was: maleic acid > acetoacetic acid > glycolic acid. The fitting results of the other two diffusion models are shown in electronic supplementary material, figure S3–S8, and electronic supplementary material, table S9 and S10. According to the $R^2$ value of the fitting results of the three models, the rate-controlling step of the acid-leaching process by the three organic acids was the surface chemical reaction step [37].

The activation energy ($E_a$) of the leaching reaction indicates the energy barrier of the process. The higher the $E_a$, the more difficult it is for the reaction to proceed. The Arrhenius equation is used to calculate the activation energy ($E_a$) of the leaching process:

$$\left. \begin{aligned} k &= Ae^{-E_a/RT} \\ \ln k &= \ln A - \frac{E_a}{RT}, \end{aligned} \right\} \tag{3.5}$$

and

where $k$ is the chemical surface reaction constant ($min^{-1}$), $A$ is the pre-exponential factor, $E_a$ is the apparent activation energy, $R$ is the universal gas constant (8.314 J K$^{-1}$ mol$^{-1}$) and $T$ is the absolute

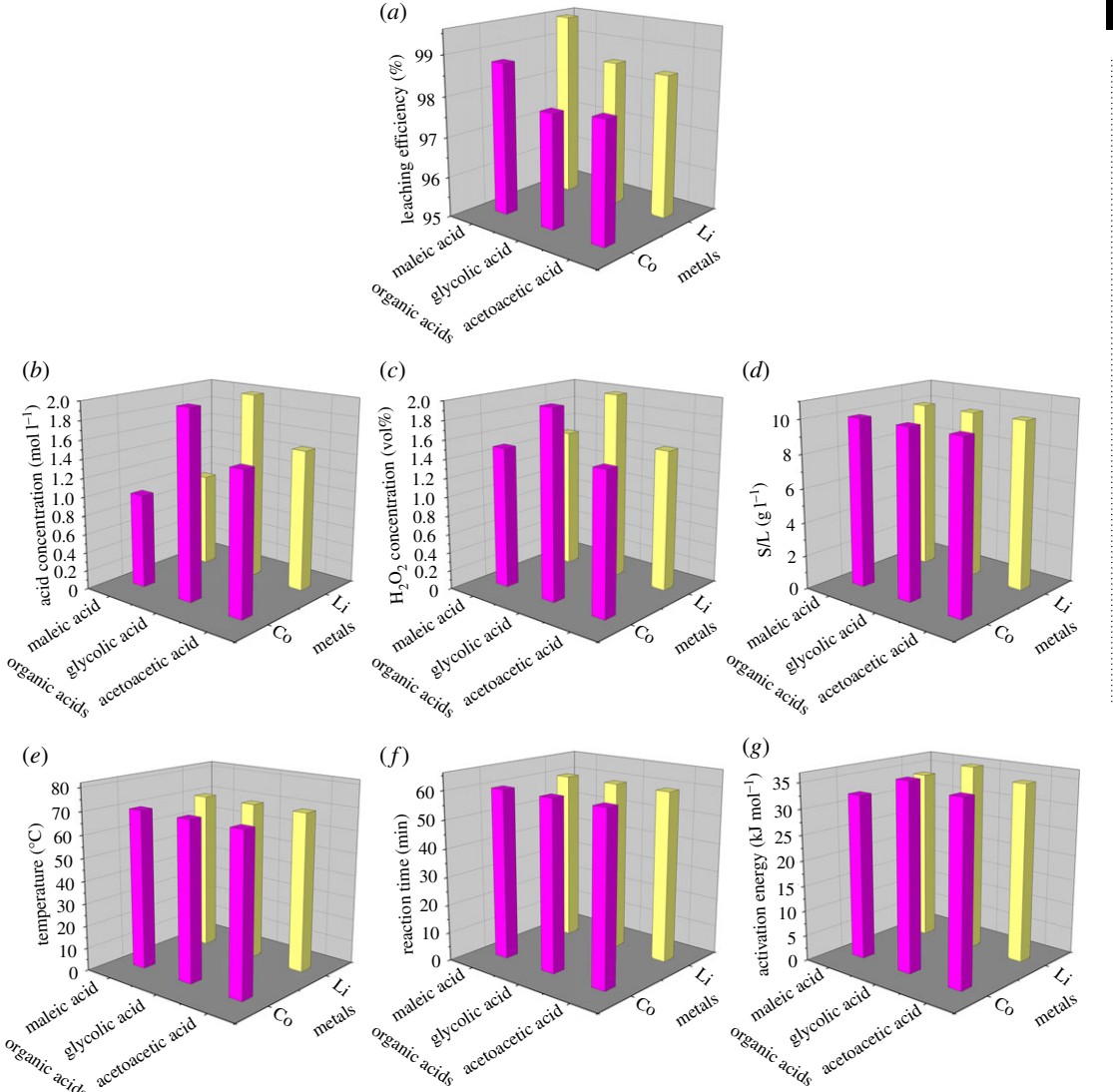

**Figure 10.** Comparison between the three organic acids in the leaching efficiency (*a*), optimal acid concentration (*b*), optimal H$_2$O$_2$ concentration (*c*), optimal S/L ratio (*d*), temperature (*e*), reaction time (*f*) and activation energy (*g*).

temperature ($K$). The plots of ln $k$ versus $1/T$ are shown in figures 7–10. The results indicated that for the maleic acid-leaching process, the activation energy of Li and Co leaching was 33.86 and 32.40 kJ mol$^{-1}$, respectively. For the glycolic acid-leaching process, the activation energy of Li and Co leaching was 36.79 and 36.60 kJ mol$^{-1}$, respectively. And, for the acetoacetic acid-leaching process, the activation energy of Li and Co leaching was 35.01 and 35.26 kJ mol$^{-1}$, respectively. The kinetic study results indicated that the rate of the leaching processes by the three organic acids was controlled by the surface chemical reaction step, and the order of energy barrier was: maleic acid < acetoacetic acid < glycolic acid. Except that, according to the properties of acids and results of thermodynamic analysis, the dissociation constant of maleic acid was the highest, while the $\Delta_r G_{m\ Maleic}$ was the lowest. Therefore, the leaching rate of metals by maleic acid was the highest, indicating that acid-leaching reaction of maleic acid was the easiest to proceed, while that of glycolic acid was the hardest.

## 3.4. Comparison of the three organic acids as the leaching agents

In order to assess and rank the availability and efficacy of the three organic acids as leaching agents, a comparison was made in the leaching efficiency, optimal leaching parameters (i.e. acid concentration, H$_2$O$_2$ concentration, S/L ratio, temperature, reaction time) and the activation energy of these acids, whose results are shown in figure 10. The leaching efficiencies of Li and Co by the three organic acids were in the following order: maleic acid > acetoacetic acid > glycolic acid. The order of the optimal

acid concentrations was: maleic acid < acetoacetic acid < glycolic acid, while that of the optimal $H_2O_2$ concentrations was: maleic acid = acetoacetic acid < glycolic acid. The order of activation energy for the three organic acids was the same as that of the optimal acid concentrations. There is no difference between the three organic acids in the optimal S/L ratios, temperatures and reaction times. Moreover, according to equations (2.2)–(2.5), the dissociation constants of acids were in the following order: maleic acid > acetoacetic acid > glycolic acid, indicating that maleic acid could provide the most $H^+$ for the leaching reactions, while that provided by glycolic acid was the least. Based on the leaching efficiencies of Co and Li by the three organic acids under the same acid concentration (figures 2a–4a), the equilibrium leaching efficiencies of metals were in the following order: maleic acid > acetoacetic acid > glycolic acid. What's more, based on the thermodynamic results, the order of $\Delta_r G_m$ of the leaching process by the three acids under the optimal temperature (70°C) was $\Delta_r G_{m\ \text{Maleic}} < \Delta_r G_{m\ \text{Acetoacetic}} < \Delta_r G_{m\ \text{Glycolic}}$ (electronic supplementary material, table S7), indicating that the extent to which the acid-leaching reactions proceeded was in the following order: maleic acid > acetoacetic acid > glycolic acid. Conclusively, among the three organic acids, the overall efficacy and availability of maleic acid as the leaching agent was the best, while those of glycolic acid was the worst.

# 4. Conclusion

Maleic, glycolic and acetoacetic acids are investigated in this study for their acid-leaching recovery abilities of metals in cathodic material from spent LIBs. When maleic acid is employed as the leaching agent, the leaching efficiency of Li reaches 99.58%, while that of Co is 98.77%. The optimal leaching condition for maleic acid is acid concentration = 1 mol l$^{-1}$, $H_2O_2$ concentration = 1.5 vol%, solid/liquid ratio = 10 g l$^{-1}$, temperature = 70°C and reaction time = 60 min. The glycolic acid-leaching efficiencies are 97.54% and 97.83% for Li and Co, respectively. The optimal leaching parameters are acid concentration = 2 mol l$^{-1}$, $H_2O_2$ concentration = 2 vol%, solid/liquid ratio = 10 g l$^{-1}$, temperature = 70°C and reaction time = 60 min. When acetoacetic acid is employed as the leaching agent, the leaching efficiency of Li is 98.62%, while that of Co is 97.99%. The optimal leaching parameters are acid concentration = 1.5 mol l$^{-1}$, $H_2O_2$ concentration = 1.5 vol%, solid/liquid ratio = 10 g l$^{-1}$, temperature = 70°C and reaction time = 60 min.

Thermodynamics and kinetics studies were employed to investigate the mechanisms of the leaching processes by the three organic acids. The Gibbs free energy of the reactions indicates that maleic acid-leaching reaction is the easiest to proceed, while the extent of the glycolic acid-leaching reaction is the hardest. The leaching processes agreed well with the shrinking-core kinetic model, and the surface chemical reaction is the rate-controlling step. Based on the comparison of the leaching efficiency, optimal leaching condition, Gibbs free energy and the activation energy, the efficacy and availability of the three organic acids is as follows: maleic acid > acetoacetic acid > glycolic acid.

Data accessibility. There are no additional data to accompany this manuscript and the electronic supplementary material. All relevant datasets are within the main body of the manuscript or the electronic supplementary material.
Authors' contributions. B.L., Q.H. and Y.S. conceived and designed the experiment. B.L., L.S., T.W. and G.W. performed the experiment. R.M.K. and F. W. analysed the data and wrote the paper. All authors gave final approval for publication.
Competing interests. We have no competing interests.
Funding. This work was supported by National Key R&D Program of China (2016YFB0100301), National Natural Science Foundation of China (21875022, U1664255).

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
