## [Reviewer comments · Royal Society Open Science]

Review History

RSOS-191061.R0 (Original submission)

Review form: Reviewer 1

Is the manuscript scientifically sound in its present form?

No

Are the interpretations and conclusions justified by the results?

No

Is the language acceptable?

Yes

Do you have any ethical concerns with this paper?

No

Have you any concerns about statistical analyses in this paper?

No

Recommendation?

Major revision is needed (please make suggestions in comments)

Comments to the Author(s)

The manuscript presents “Leaching of Valuable metals from spent LIBs: Optimization, Thermodynamics and kinetics of leaching”. In this study, the authors are using the organic acids which have been not used by other researchers. Although, the work has the scientific novelty in terms of using new leaching agents to leach metals from spent LIBs, the thermodynamical interpretation and kinetic analysis should be more comprehensive to understand the leaching behavior of metals by organic acids such as maleic, glycolic and acetoacetic acid. The same concentration of acids should be employed to compare the leachability to dissolve metals, but the effect of different concentrations (1.0 M for maleic, 2.0 M for glycolic, and 1.5 M for acetoacetic acid) has been compared under the optimized leaching conditions. The comprehensive interpretation of leaching should be conducted with the dissociation constant of acids, activation energy, Gibbs free energy etc.

Review form: Reviewer 2

Is the manuscript scientifically sound in its present form?

Yes

Are the interpretations and conclusions justified by the results?

Yes

Is the language acceptable?

Yes

Do you have any ethical concerns with this paper?

No

Have you any concerns about statistical analyses in this paper?

Yes

Recommendation?

Major revision is needed (please make suggestions in comments)

Comments to the Author(s)

In this study, Maleic, glycolic and acetoacetic acids were respectively adopted by the authors during the reductive leaching process for the recycling of valuable metals from spent LIBs batteries. Despite sound results can be obtained, several issues should be taken due consideration before its publication in this journal.

1. The first issue, also the most important one, concerning this manuscript is the logic and arrangement of this text, the introduction part cannot present a clear cue for potential readers and there are also several errors (like line 59, page 1 “i.e. ~500 tons”; line 9, page 2 “hydrometallurgical [12], and biometallurgical methods [13]” I think the bio-leaching method can be concluded in hydrometallurgical method.). The most recent developments involved with leaching process are also adopted like phosphoric acid as inorganic acid (e.g. Waste Management 85 (2019) 175–185; Waste Management 80 (2018) 198–210; Journal of Hazardous Materials 326

(2017) 77–86) and organic acids (Separation and Purification Technology 210 (2019) 690–697; ACS Sustainable Chem. Eng. 2017, 5, 714–721; Waste Manag. Res., 2018, 36(2), 113–120), there are may be also other newly developed acid leaching systems, please update them and give potential readers more precise and clear statement and discussion in second paragraph 2 of page 2. In conclusion, the introduction part should be significant revision.

2. The second issue is the results and discussion section, the 4.1, 4.2 and 4.2.1 sections should be mainly discussed in the experimental section instead of this section. In section 4.3 I do not think it is a thermodynamic analysis, please pay due attention to revise this section;

3. The novelty of this manuscript derives from the adoption of different kinds of organic acid. However, as we all known, organic acids are usually expensive than those of inorganic acids, so is it profitable for the adoption of these expensive acids instead if inorganic ones? What about their environmental impacts regarding TOC and other organic hazardous substances?

Review form: Reviewer 3

Is the manuscript scientifically sound in its present form?

No

Are the interpretations and conclusions justified by the results?

No

Is the language acceptable?

No

Do you have any ethical concerns with this paper?

No

Have you any concerns about statistical analyses in this paper?

No

Recommendation?

Reject

Comments to the Author(s)

The article entitle "Maleic, glycolic and acetoacetic acids-leaching for recovery of valuable metals from spent LIBs: Leaching parameters, thermodynamics and kinetics" was carefully reviewed. The overall present data is not good and not comparable with literature. In the present form the article is not acceptable for the publication.

Decision letter (RSOS-191061.R0)

08-Jul-2019

Dear Dr Huang:

Title: Maleic, glycolic and acetoacetic acids-leaching for recovery of valuable metals from spent LIBs: Leaching parameters, thermodynamics and kinetics
Manuscript ID: RSOS-191061

The editor assigned to your manuscript has now received comments from reviewers. We would like you to revise your paper in accordance with the referee and Subject Editor suggestions which can be found below (not including confidential reports to the Editor). Please note this decision does not guarantee eventual acceptance.

Please submit your revised paper before 31-Jul-2019. Please note that the revision deadline will expire at 00.00am on this date. If we do not hear from you within this time then it will be assumed that the paper has been withdrawn. In exceptional circumstances, extensions may be possible if agreed with the Editorial Office in advance. We do not allow multiple rounds of revision so we urge you to make every effort to fully address all of the comments at this stage. If deemed necessary by the Editors, your manuscript will be sent back to one or more of the original reviewers for assessment. If the original reviewers are not available we may invite new reviewers.

- Acknowledgements

RSC Associate Editor:
Comments to the Author:
(There are no comments.)

RSC Subject Editor:
Comments to the Author:
(There are no comments.)

Reviewers' Comments to Author:
Reviewer: 1

Comments to the Author(s)

The manuscript presents "Leaching of Valuable metals from spent LIBs: Optimization, Thermodynamics and kinetics of leaching". In this study, the authors are using the organic acids which have been not used by other researchers. Although, the work has the scientific novelty in terms of using new leaching agents to leach metals from spent LIBs, the thermodynamical interpretation and kinetic analysis should be more comprehensive to understand the leaching behavior of metals by organic acids such as maleic, glycolic and acetoacetic acid. The same concentration of acids should be employed to compare the leachability to dissolve metals, but the effect of different concentrations (1.0 M for maleic, 2.0 M for glycolic, and 1.5 M for acetoacetic acid) has been compared under the optimized leaching conditions. The comprehensive interpretation of leaching should be conducted with the dissociation constant of acids, activation energy, Gibbs free energy etc.

Reviewer: 2

Comments to the Author(s)

In this study, Maleic, glycolic and acetoacetic acids were respectively adopted by the authors during the reductive leaching process for the recycling of valuable metals from spent LIBs batteries. Despite sound results can be obtained, several issues should be taken due consideration before its publication in this journal.

1. The first issue, also the most important one, concerning this manuscript is the logic and arrangement of this text, the introduction part cannot present a clear cue for potential readers and there are also several errors (like line 59, page 1 "i.e. ~500 tons"; line 9, page 2 "hydrometallurgical [12], and biometallurgical methods [13]" I think the bio-leaching method can be concluded in hydrometallurgical method.). The most recent developments involved with leaching process are also adopted like phosphoric acid as inorganic acid (e.g. Waste Management 85 (2019) 175–185; Waste Management 80 (2018) 198–210; Journal of Hazardous Materials 326 (2017) 77–86) and organic acids (Separation and Purification Technology 210 (2019) 690–697; ACS Sustainable Chem. Eng. 2017, 5, 714–721; Waste Manag. Res., 2018, 36(2), 113–120), there are may be also other newly developed acid leaching systems, please update them and give potential readers more precise and clear statement and discussion in second paragraph 2 of page 2. In conclusion, the introduction part should be significant revision.
2. The second issue is the results and discussion section, the 4.1, 4.2 and 4.2.1 sections should be mainly discussed in the experimental section instead of this section. In section 4.3 I do not think it is a thermodynamic analysis, please pay due attention to revise this section;
3. The novelty of this manuscript derives from the adoption of different kinds of organic acid. However, as we all known, organic acids are usually expensive than those of inorganic acids, so

is it profitable for the adoption of these expensive acids instead of inorganic ones? What about their environmental impacts regarding TOC and other organic hazardous substances?

Reviewer: 3

Comments to the Author(s)

The article entitled "Maleic, glycolic and acetoacetic acids-leaching for recovery of valuable metals from spent LIBs: Leaching parameters, thermodynamics and kinetics" was carefully reviewed. The overall present data is not good and not comparable with literature. In the present form the article is not acceptable for the publication.

Author's Response to Decision Letter for (RSOS-191061.R0)

See Appendix A.

Decision letter (RSOS-191061.R1)

12-Aug-2019

Dear Dr Huang:

Title: Maleic, glycolic and acetoacetic acids-leaching for recovery of valuable metals from spent LIBs: Leaching parameters, thermodynamics and kinetics
Manuscript ID: RSOS-191061.R1

It is a pleasure to accept your manuscript in its current form for publication in Royal Society Open Science. The chemistry content of Royal Society Open Science is published in collaboration with the Royal Society of Chemistry.

RSC Associate Editor
Comments to the Author:
(There are no comments.)

Reviewer(s)' Comments to Author:

Appendix A

Response to the Comments of the Editors and Referees (Manuscript No. RSOS-191061)

Dear editor and reviewer,

We would like to thank you for giving us a chance to revise the paper *No. RSOS-191061*, and also thank the reviewers for giving us constructive suggestions. Those comments are all valuable and very helpful for revising and improving our paper, as well as the important guiding significance to our researches. We have studied comments carefully and have made thorough corrections which we hope to meet the requirement. The differences between the revision and the original manuscript are present in the form of ‘tracked changes’ and uploaded as ‘Responds to referees’ in the online submission system. The point-by-point answers to the comments and suggestions were listed as below.

To Reviewer #1

Question 1:

The manuscript presents “Leaching of Valuable metals from spent LIBs: Optimization, Thermodynamics and kinetics of leaching”. In this study, the authors are using the organic acids which have been not used by other researchers. Although, the work has the scientific novelty in terms of using new leaching agents to leach metals from spent LIBs, the thermodynamical interpretation and kinetic analysis should be more comprehensive to understand the leaching behavior of metals by organic acids such as maleic, glycolic and acetoacetic acid. The same concentration of acids should be employed to compare the leachability to dissolve metals, but the effect of different concentrations (1.0 M for maleic, 2.0 M for glycolic, and 1.5 M for acetoacetic acid) has been compared under the optimized leaching conditions. The comprehensive interpretation of leaching should be conducted with the dissociation constant of acids, activation energy, Gibbs free energy etc.

Answer: We apologize for our negligence. About the suggestions in this question, we have made some improvement in the revised manuscript.

(1) In order to make the thermodynamic interpretation and kinetic analysis more comprehensive, we have rewritten a lot of these two parts (Section 4.2 and 4.3). In Section 4.2, we have added the purpose of the thermodynamic study, with the meaning of the two thermodynamic parameters (formation energy and Gibbs free energy). We have made the reaction process more specific, as well as rearrange the content of this section to make it more comprehensive. At the end of the section, we have added a summary about the information given by the thermodynamic results.

In Section 4.3, we have also added the purpose of the kinetic study, with the meaning of the kinetic parameter (activation energy). We have added more specific information about how to obtain the rate-controlling step according to the equation fitting results. At the end of the section, we have added a summary about the information given by the analysis of thermodynamic results.

(2) We have added the data about Gibbs free energy in the revised manuscript, as can be shown in Section 4.2 as well as Table S7. We have also added the relevant contents in the section of *Summary, Introduction and Conclusions*

(3) In Section 4.3, we have added the interpretations of leaching rate with dissociation constant, Gibbs free energy and activation energy. In Section 4.4, we have compared the leaching efficiencies of metals by the three acids under the same acid concentrations. What's moer, we have compared the dissociation constant of acids, activation energy and Gibbs free energy of

the leaching reactions, in order to comprehensively compare the leachability of different organic acids.

Question 2:

Some comments are given to improve the manuscript as follows;

- Stoichiometric amount of acids to leach metal components from spent LIBs?

Answer: We apologize for our negligence. We have added the chemical equations of the three acid-leaching reactions as Eq (7) to (9) in the revised manuscript to illustrate the stoichiometric amount of acids to leach metal components.

Question 3:

- In pages 6, lines 11–16, the authors mentioned “As the H₂O₂ concentration further rose to 2 and 2.5 vol %, the leaching efficiency of Li and Co was maintained at a stable level. The reason for this phenomenon might be caused when the H₂O₂ concentration was high, the excessive H₂O₂ was unstable and decomposed into H₂O and O₂ “. But, Co of LiCoO₂ is mostly leached in the presence of 1.5 vol% H₂O₂ and Co of Co₂O₃ present in the leaching residue is not leached. The reason for no further leaching of Co with H₂O₂ is not due to the instability of H₂O₂, but is due to the complete leaching of Co which can be leached with the presence of H₂O₂.

Answer: We apologize for our negligence. We have rewritten the interpretation of this experimental phenomenon in the Section 4.1.1 as the suggestion. And we have removed the original Reference [31].

Question 4:

- In pages 6, lines 23–24, “This discovery could result from when the S/L ratio was high, the available surface area decreased significantly, resulting in lower leaching efficiency”. The reviewer does not agree to the authors’ interpretation. It can be described to the shortage of acid to leach metals with increase of pulp density.

Answer: We apologize for our negligence. We have removed the original interpretation of this experimental phenomenon in the Section 4.1.1 as the suggestion. And we have removed the original Reference [32].

Question 5:

- In pages 6, lines 31–34, “As the temperature further reached 80 and 90 °C, the leaching efficiency of Li and Co was maintained at a stable level. The reason for this respond might be due to leaching reaction is being an endothermic reaction [33].” The reviewer does not agree to the authors’ interpretation.

Answer: We apologize for our negligence. We have rewritten the interpretation of this experimental phenomenon in the Section 4.1.1 as the suggestion. And we have removed the original Reference [33].

Question 6:

- The formation energy could be expressed as SI unit?

Answer: We apologize for our negligence. We have changed the unit of formation energy into SI unit in the Section 4.2 Table S6 as the suggestion.

Question 7:

- Leaching rate (kinetics, not efficiency) should be interpreted with dissociation constant of acids and Gibbs free energy and activation energy to compare the leachability of different organic acids.

Answer: We apologize for our negligence. We have added the data about Gibbs free energy in the revised manuscript, as can be shown in Section 4.2 as well as Table S7. We have also added the relevant contents in the section of *Summary, Introduction* and *Conclusions*. In Section 4.3, we have added the interpretations of leaching rate with dissociation constant, Gibbs free energy and activation energy. In Section 4.4, we have compared the dissociation constant of acids, activation energy and Gibbs free energy of the leaching reactions, in order to comprehensively compare the leachability of different organic acids.

To Reviewer #2**Question 1:**

In this study, Maleic, glycolic and acetoacetic acids were respectively adopted by the authors during the reductive leaching process for the recycling of valuable metals from spent LIBs batteries. Despite sound results can be obtained, several issues should be taken due consideration before its publication in this journal.

1. The first issue, also the most important one, concerning this manuscript is the logic and arrangement of this text, the introduction part cannot present a clear cue for potential readers and there are also several errors (like line 59, page 1 “i.e. ~500 tons”; line 9, page 2 “hydrometallurgical [12], and biometallurgical methods [13]” I think the bio-leaching method can be concluded in hydrometallurgical method.). The most recent developments involved with leaching process are also adopted like phosphoric acid as inorganic acid (e.g. *Waste Management* 85 (2019) 175–185; *Waste Management* 80 (2018) 198–210; *Journal of Hazardous Materials* 326 (2017) 77–86) and organic acids (*Separation and Purification Technology* 210 (2019) 690–697; *ACS Sustainable Chem. Eng.* 2017, 5, 714–721; *Waste Manag. Res.*, 2018, 36(2), 113-120), there are may be also other newly developed acid leaching systems, please update them and give potential readers more precise and clear statement and discussion in second paragraph 2 of page 2. In conclusion, the introduction part should be significant revision.

Answer: We apologize for our negligence. About the suggestions in this question, we have made some improvement in the revised manuscript.

- (1) We have rewritten a lot of the section of *Introduction* to make it more logic and comprehensive, and we have also added a new Reference [11]. We have removed the original References [5] and [6] to meet the requirement.
- (2) We have corrected the errors in the section of *Introduction* mentioned here. The predicted LIBs number in 2020 has been corrected. We have also concluded the original *hydrometallurgical and biometallurgical methods* into the new *hydrometallurgical method*.
- (3) We have added the references mentioned in this question (Reference [18][19][20][35][34][28]);
- (4) We have added some other references about newly developed acid leaching system (Reference [16][23][25][26][36]).

Question 2:

The second issue is the results and discussion section, the 4.1, 4.2 and 4.2.1 sections should be

mainly discussed in the experimental section instead of this section. In section 4.3 I do not think it is a thermodynamic analysis, please pay due attention to revise this section;

Answer: We apologize for our negligence. About the suggestions in this question, we have made some improvement in the revised manuscript.

- (1) We have rearranged the contents about the spent material characterization, organic properties, and orthogonal experiment in the section of *Experimental Procedure* (Section 3.2.1, 3.2.2 and 3.2.3). Moreover, we have rearranged the contents in the Section 3.2 to make it more comprehensive.
- (2) In Section 4.3, we have added the calculation of Gibbs free energy of the leaching reactions. The formation energy of the leaching products in the original manuscript could provide basis for the calculation of Gibbs free energy which is an important thermodynamic parameter of the leaching reactions. We hope that this correction could make the contents appropriate as the thermodynamic analysis.

Question 3:

The novelty of this manuscript derives from the adoption of different kinds of organic acid. However, as we all known, organic acids are usually expensive than those of inorganic acids, so is it profitable for the adoption of these expensive acids instead if inorganic ones? What about their environmental impacts regarding TOC and other organic hazardous substances?

Answer: We apologize for our negligence. We have noticed that in some literature, the organic acids leachate of spent LIBs can be directly used as the material to resynthesize cathodic material of LIBs through sol-gel method, without any intermediate step or loss of materials. Therefore, a closed-loop process of recovery-resynthesis can be achieved when organic acids are employed as the leaching agents. This process can not only reduce the recovery cost, but also minimize the emission of TOC and other organic hazardous substances. Therefore, we think that the organic acids are profitable in the acid-leaching process. We have added the relevant contents in the section of *Introduction*, and we plan to conduct relevant study in our subsequent research.

To Reviewer #3

Question:

The article entitled "Maleic, glycolic and acetoacetic acids-leaching for recovery of valuable metals from spent LIBs: Leaching parameters, thermodynamics and kinetics" was carefully reviewed. The overall present data is not good and not comparable with literature. In the present form the article is not acceptable for the publication.

Answer: We apologize for our negligence and confusion that we have caused. And we would like to thank you for your effort on reviewing our manuscript.

In this paper, we report on the availability of maleic, glycolic and acetoacetic acids for recovery of lithium and cobalt from cathode of spent lithium batteries. We report on the effects of leaching parameters (i.e. acid concentration, reductant dosage, solid/liquid ratio, temperature and reaction time) on the leaching efficiencies of these organic acids. We think that there is sufficient novelty in this manuscript, which could also provide valuable information for the recycling practice for the spent LIBs.

The novelty lies in the following aspects:

- (a) maleic, glycolic and acetoacetic acids are employed for the acid-leaching recovery process for

the cathode of spent lithium ion batteries, and there is limited information about the availability of these organic acids;

(b) we conducted thermodynamic and kinetic studies to investigate the leaching reaction mechanisms, the thermodynamic and kinetic behavior of the leaching system, as well as the feasibility and extent of the leaching reactions. Especially in the thermodynamic part, we used Materials Studios software to calculate the formation energy of the possible leaching products, in order to recognize the thermodynamic favorable molecular structure of the maleic acid-leaching products. Based on the results, in the revised manuscript, the Gibbs free energy of the leaching reactions was calculated, where there is limited information in the literature. For the kinetic study, we have investigated the shrinking-core kinetic model for the leaching reactions, as well as calculated the activation energy. The thermodynamic and kinetic investigation could help a lot to understand the acid-leaching mechanisms;

(c) we compared and ranked the overall efficacy of the three organic acids. In the revised manuscript, the comparison is not only based on the leaching efficiencies, but also based on all the process conditions and thermodynamic/kinetic parameters, including dissociation constants of the acids, reactant concentrations, solid/liquid ratio, Gibbs free energy, activation energy. The rank of the overall availability of the three organic acids could provide important information for the recycling technologies of spent LIBs. Moreover, the comparison method of the availability of organic acids in the acid-leaching process has potential to be further investigated and employed.

We hope that our manuscript could meet the requirement for publication in this journal. Thank you so much for your efforts.

Other revisions

- (a) Because of the rearrangement of the contents in the manuscript, some numbers of equations, tables and references have been changed in the revised manuscript, as well as in the revised supplementary materials.
- (b) To avoid language problems, the English text in the manuscript has been polished by a native English-speaking expert (i.e. the seventh author of the paper) again.

That is the list of changes for the revised manuscript. We would like to thank you for giving us a chance to revise the paper, and also thank the reviewers for giving us constructive suggestions. If there is any unaccomplished matter, please contact us.

Best regards!

Yours sincerely,

Borui Liu

Qing Huang

Yuefeng Su

Liuye Sun

Tong Wu

Guange Wang

Ryan M. Kelly

Feng Wu